# Dalbavancin as Suppressive Therapy for Implant-Associated Osteoarticular Infections

**DOI:** 10.3390/antibiotics14111171

**Published:** 2025-11-20

**Authors:** Rosa Escudero-Sanchez, Laura Morata, Luis Buzón, Sofia de la Villa, Alicia Rico, María José Nuñez Orantos, Laura Guio Carrion, María Tasias Pitarch, Jose Luis del Pozo, José M. Barbero, Joan Gómez-Junyent, María José García Pais, Pablo Bachiller Luque, Francisco Javier Martínez Marcos, Javier Cobo

**Affiliations:** 1Infectious Disease Department, Ramon y Cajal University Hospital, 28034 Madrid, Spain; 2CIBERINFEC (Network Biomedical Research Center in Infectious Disease), 28034 Madrid, Spain; 3IRYCIS (Ramón y Cajal Institute of Health Research), 28034 Madrid, Spain; 4Internal Medicine Department, Clinic University Hospital, 08041 Barcelona, Spain; 5Internal Medicine Department, Burgos University Hospital, 09006 Burgos, Spain; 6Clinical Microbiology and Infectious Diseases Department, Gregorio Marañón University Hospital, 28007 Madrid, Spain; 7IISGM (Gregorio Marañón Health Research Institute), 28009 Madrid, Spain; 8Microbiology Department, La Paz University Hospital, 28046 Madrid, Spain; 9Internal Medicine Department, Clínico San Carlos University Hospital, 28040 Madrid, Spain; 10Infectious Disease Department, Biobizkaia Biomedical Research Institute, Cruces University Hospital, 48903 Vizcaya, Spain; 11Infectious Disease Department, Hospital La Fe, 46026 Valencia, Spain; 12Infectious Diseases Division, Department of Microbiology, Clínica Universidad de Navarra, 31008 Pamplona, Spain; 13Instituto de Investigación Sanitaria de Navarra (IdiSNA), 31008 Pamplona, Spain; 14Internal Medicine Department, Príncipe Asturias University Hospital, 28805 Alcalá de Henares, Spain; 15Infectious Disease Department, Hospital del Mar, 08003 Barcelona, Spain; 16Research Institute, UPF (Universitat Pompeu Fabra), 08003 Barcelona, Spain; 17Internal Medicine Department, Lugo Hospital, 27003 Lugo, Spain; 18Santiago de Compostela Research Health Institute, 27003 Santiago de Compostela, Spain; 19Internal Medicine Department, Segovia Hospital, 40002 Segovia, Spain; 20Infectious Disease Department, Juan Ramón Jiménez Hospital, 21005 Huelva, Spain

**Keywords:** suppressive antibiotic treatment, osteoarticular implant-associated infection, prosthetic joint infection, dalbavancin

## Abstract

**Introduction**: Suppressive antibiotic therapy (SAT) is a therapeutic alternative for complex infections where a cure is considered unlikely or impossible. SAT involves the prolonged, often indefinite, administration of antibiotics, typically given orally, to control symptoms. However, the increasing incidence of multidrug-resistant microorganisms limits the availability of oral options. Dalbavancin is a parenteral antibiotic with broad coverage against Gram-positive bacteria that offers the advantage of an extended dosing interval. The aim of this study was to describe the characteristics and clinical outcomes of patients with implant-associated osteoarticular infections receiving dalbavancin as SAT. A secondary objective was to identify factors associated with SAT failure with dalbavancin. **Materials and Methods**: We conducted a multicentre, observational study with retrospective recruitment of patients treated with dalbavancin as (SAT) for complex implant-associated osteoarticular infections, in which curative surgery was either not feasible or insufficient. Cohort characteristics were described, and variables associated with SAT failure under dalbavancin treatment were analysed. **Results**: A total of 43 patients received dalbavancin as SAT. The most frequent indication was prosthetic joint infection (38 [88.4%]). A total of 28 patients (65.1%) had chronic infections; the remaining cases were acute infections that had failed conservative management. Nine different dosing regimens of dalbavancin were used. Dalbavancin provided adequate symptomatic control in 32 patients (74.4%) over a follow-up period of 836.5 days (IQR 402–1288.5). The antibiotic was well tolerated; only one adverse effect was reported in a patient. Three patients developed resistance during treatment, which accounted for SAT failure. **Conclusions**: Dalbavancin is shown to be a safe and convenient alternative for SAT for orthopaedic implant infection. Although the development of resistance was infrequent, it can occur and should be monitored.

## 1. Introduction

Suppressive antibiotic therapy (SAT) represents a therapeutic alternative for patients with complex infections in whom the surgery required for cure cannot be performed or is considered insufficient to eradicate the infection. SAT is also liberally used for patients at risk of failure even though the surgery performed is potentially curative [1,2]. Thus, SAT involves prolonged, usually indefinite, administration of oral antibiotics with the aim of alleviating symptoms and/or preventing progression of the infection [3]. Although it may be applied in various types of complex infections, generally those associated with implants, its main use lies in osteoarticular infections.

The increasing incidence of multidrug-resistant microorganisms, drug–drug interactions and adherence, often limits of the use of oral antibiotics. The recent introduction of dalbavancin and other long-acting parenteral antibiotics—allowing weekly, biweekly, or even monthly dosing—with excellent activity against Gram-positive bacteria (the predominant pathogens in these infections) has raised interest in their potential utility for SAT [4].

Dalbavancin is generally used off-label, beyond its approved indication for skin and soft tissue infections, particularly in osteoarticular and cardiovascular infections [5,6,7,8,9]. To date, only a few anecdotal reports have described dalbavancin use as SAT [6,8,10]. The development of resistance during dalbavancin therapy is rare, although it has been described in certain reports involving biofilm-associated infections [11,12]. Due to the small sample sizes of existing studies, several questions remain unanswered regarding dalbavancin in this context, including: (1) the most appropriate dosing regimens; (2) its safety during long-term use; and (3) the risk of resistance development.

In the present study, we evaluate the real-life effectiveness of dalbavancin as SAT in implant-associated osteoarticular infections.

## 2. Results

Forty-three patients from 13 hospitals received dalbavancin in SAT with for osteoarticular infections. The median age was 78 years (IQR 67–83), and the majority were women (26; 60.5%). The Charlson comorbidity index was 2 (IQR 0–3). The most frequent comorbidities were conditions associated with immunosuppression (10; 23.3%), solid or hematologic malignancies (10; 23.3%), diabetes mellitus (7; 16.3%), and chronic kidney disease (6; 14.0%). Seven patients had a history of allergy to at least one antibiotic class (four of them were allergic to β-lactams).

The most common indication for SAT was prosthetic joint infection (38; 88.4%), followed by osteosynthesis material infection (3; 7.0%) and vertebral instrumentation (2; 4.7%). Chronic infections were observed in 28 patients (65.1%). All three osteosynthesis material infections and one of the vertebral instrumentations were chronic. No surgical procedure was performed in 18 patients (41.9%). Among those who underwent surgery, DAIR (debridement, antibiotics, and implant retention) was the most frequent approach (21; 48.8%). SAT was used in 11 acute infections (25.6%) managed with DAIR due to early failure. Median time from implant placement to infection diagnosis was 367 days (IQR 6–964).

SAT was initiated specifically due to treatment failure, even in patients whose surgical procedure had initially been considered adequate for their infection (13; 30.2%). *Staphylococcus epidermidis* was the most frequently isolated microorganism (19; 47.5%). Table 1 shows the cohort characteristics and Figure 1 describes presenting symptoms at SAT initiation and at the time of evaluation.

Reasons for selecting dalbavancin over other options included improved adherence (48.8%), toxicity of other antibiotics (41.9%), and resistance to alternative therapies (27.9%) (several options are possible). The first dalbavancin dose was most often administered in the outpatient setting (25; 58.1%). In our cohort, nine different dosing schedules were prescribed. Administration every two weeks was prescribed in 20 patients (46.7%). Table 2 summarises dalbavancin administration characteristics.

The overall follow-up period of the cohort was 617 days (IQR 305–1143). During this time, 11 patients (25.6%) experienced treatment failure as defined in the study. The median time to failure was 313 days (IQR 60–576). The most common cause of failure was persistent infection (10; 23.3%), of which 3 cases (7.0% of the cohort; 27.3% of failures) were associated with the development of resistance. Five patients (11.6%) failed due to the persistence of the same microorganism as in the initial infection. Median time to failure did not differ significantly between patients with resistance-related failures and those with other causes (276 days [IQR 27–1053] vs. 408.5 days [IQR 90–548.5], respectively; *p* = 0.36). At the time of analysis, 22 patients remained on dalbavancin SAT (52.4%). Table 3 shows the patients who developed dalbavancin resistance. Appendix A describes failures.

Only one patient (2.3%) developed an adverse event, described as liver enzyme alteration, although the causality with dalbavancin was uncertain. According to our definition, it was also classified as a failure. During follow-up, 5 patients (11.6%) died, none due to the osteoarticular infection (time to death: 1444 days [IQR 664–1881]). Table 4 summarises follow-up events.

No associations were identified between the variables analysed and SAT failure with dalbavancin (Appendix A).

## 3. Discussion

Our study represents the largest cohort of patients treated with dalbavancin on SAT for osteoarticular infections, with the highest cumulative doses and longest follow-up reported to date. Three-quarters of patients achieved symptomatic control for more than two years, and the drug was very well tolerated.

Achieving symptomatic control with SAT represents a therapeutic challenge in daily clinical practice. Orthopaedic procedures are increasingly complex and performed in patients with greater comorbidity, which complicates antibiotic selection. Multiple factors influence this decision: infections caused by multidrug-resistant microorganisms, drug–drug interactions in patients with comorbidities, bioavailability of antibiotics, polypharmacy, and ensure adherence. The antibiotic selected for SAT should target the causative microorganism, have an adequate safety profile, minimal drug interactions, allow monitoring by a specialist, and be convenient to administer. Therefore, choosing the appropriate antibiotic for SAT represents a major challenge for infectious disease specialists [2,13,14].

The pharmacokinetic properties of dalbavancin make it an attractive option for infections caused by Gram-positive cocci. Although it is marketed for skin and soft tissue infections, increasing evidence supports its benefit in other off-label indications [15].

The management of osteoarticular infection differs from that of infections in other organs due to the difficulty of achieving concentrations above the MIC in bone tissue, which may contribute to therapeutic failure if the antibiotic is not adequately selected. Dunne et al. reported bone concentrations of 6.3 μg/g and 4.1 μg/g at 12 h and 2 weeks, respectively, following a single 1000 mg dose of dalbavancin [16]. Another important factor is the biofilm, which plays a critical role in the implant-associated failures. In vitro studies showed the dalbavancin capacity to inhibit biofilm formation by Gram-positive cocci, although its activity against in established biofilms is more modest [17].

Most of the published studies on dalbavancin use for osteoarticular infections describe treatment durations of 6–12 weeks depending on infection site and surgical management, with prolonged use limited to isolated case reports [18]. In osteoarticular infections, dalbavancin has mainly been used as a step-down therapy to replace inpatient intravenous antibiotics with an outpatient alternative, although some evidence also supports its use as empirical therapy [19].

Regarding dalbavancin as SAT for osteoarticular infection, (OAI) real-world evidence remains scarce. Lafon-Desmuers et al. [20] reported a cohort of 14 patients with chronic PJI treated with dalbavancin SAT, with 10 months of follow-up. Three patients experienced treatment failure, mainly due to superinfections with microorganisms intrinsically resistant to dalbavancin. Patients typically started with 1500 mg on days 1 and 15, followed by individualised dosing. The median number of doses was 4, with a median interval of 57 days. Dalbavancin levels remained above 4 mg/L in nearly all patients (97.9%) and above 10 mg/L in two-thirds (69%). Beyond this, evidence in the real world is limited to isolated case reports [8,10,21,22]. In the present series, dalbavancin achieved symptomatic control in 74.4% of patients during more than two years of follow-up. These results are consistent with those we previously published in a large cohort of SAT [3]. Long-term data on dalbavancin are limited to case reports and implant-associated endovascular infections, usually with treatment durations not exceeding 12 weeks. Ruiz-Sancho et al. described a cohort where dalbavancin was administrated for a knee prosthetic joint infection (two patients) and vascular infection (six patients) [21]. Patients received a median of 29 doses of dalbavancin, with dosing intervals adjusted according to clinical evolution and tolerability.

Good results with prolonged administration of dalbavancin were reported in the study by Morath et al. [22]. They described a cohort of 13 patients with ventricular assist device–associated infections who received dalbavancin every 6 weeks (1500 mg on day 1 and day 8, with the cycle repeated every 42 days). The study found that dalbavancin plasma concentrations remained above 8 mg/L throughout the 6-week dosing interval.

There is still no agreement on the optimal dosing interval for dalbavancin in SAT. Following a single dose of 1500 mg, plasma concentrations remain above efficacy thresholds (>4–8 mg/L) for several weeks [20,22,23,24]. Obviously, the 1500 mg dose would allow for longer dosing intervals, which would be more convenient for the patient and consume fewer resources. In our study, dosing was determined at the discretion of the treating physician, which is reflected in the heterogeneity of regimens within our cohort. The availability of real-time pharmacokinetic data would be of great interest to guide dosing adjustments and optimise administration intervals.

Resistance was not assessed in pivotal clinical trials for skin and soft tissue infections and has only rarely been reported in the literature [11,25]. Nevertheless, resistance to dalbavancin, while uncommon, emerged as a cause of treatment failure. In our cohort, the three patients who developed dalbavancin resistance were receiving high doses of dalbavancin at short intervals, so it does not appear that resistance developed due to insufficient dosing. Notably in two of the cases, baseline MICs to glycopeptides were elevated, which may have contributed to develop the resistance.

No risk factors associated with SAT failure were identified, although nine out of eleven cases involved knee PJI. Information regarding the higher failure rate of SAT in the knee compared with the hip is limited. Burr R. et al. suggested that the role of soft tissues in the hip, as opposed to the knee, may favour wound healing and therefore reduce the risk of prosthetic infection [26].

Dalbavancin is generally well tolerated, as shown in previous studies, although most experience relates to short-term use [27,28]. In long-term SAT, Morath B. et al. reported transaminase elevations in 30.7% of the patients, with leaded to discontinuation in only one case [22]. In our study, one patient discontinued dalbavancin due to increased liver enzymes, although an alternative cause (possible cardiac amyloidosis) could not be excluded. This case was considered a failure according to our predefined criteria.

Among the limitations of dalbavancin, intravenous administration requires venous access. However, the risk of phlebitis appears like other intravenous antibiotic administration, and permanent access is not usually necessary. Recently, subcutaneous administration of dalbavancin has been reported with promising results [29].

The main limitation of our study is the relatively small sample size, although it is the largest cohort describing dalbavancin on SAT for osteoarticular infections. Results should be interpreted as real-world data in a highly complex population, without direct comparison to other antibiotics within the same study. The homogeneity in the definitions of suppressive treatment and failure across the cohort strengthens the study, while also reflecting real-world use of dalbavancin in this clinical setting.

In summary, our cohort confirms that dalbavancin is useful and convenient for SAT in implant associated osteoarticular infections. The rate of resistance, although rare, can be one of the causes of failure.

## 4. Materials and Methods

We conducted a national multicentre observational study with retrospective recruitment of patients receiving dalbavancin as SAT for osteoarticular implant associated infections, from the commercialization of dalbavancin until the Research Ethics Committee’s approval. Participation was offered to investigators in the Spanish Osteoarticular Infection Group (GEIO) of the Spanish Society of Infectious Diseases and Clinical Microbiology (SEIMC).

We hypotheses that dalbavancin is effective and safe for use as a SAT. The primary objective was to describe the characteristics of patients treated with dalbavancin as a SAT strategy and to assess its efficacy and safety, reported in the rate of adverse events and the development of resistance during dalbavancin treatment, according EUCAST breakpoint.

Failure was defined as: (1) persistence of infection, (2) new osteoarticular infection, (3) discontinuation of dalbavancin due to toxicity, (4) need for further surgery, or (5) infection-related death. Therapeutic success was defined as the absence of these failure criteria.

### 4.1. Inclusion Criteria

Eligible patients were adults who received dalbavancin as SAT for implant associated osteoarticular infections in which medical-surgical management had not been performed or was deemed insufficient to achieve cure [10], with a minimum follow-up of six months under SAT. Patients whose initial surgery was adequate according to clinical guidelines could be included if they experienced treatment failure and were subsequently managed without further surgery.

The decision to use dalbavancin as suppressive therapy, as well as the prescribed dosage, was made by the treating physician prior to study initiation.

### 4.2. Exclusion Criteria

Patients treated with dalbavancin with curative intent or with a predefined end date were excluded.

### 4.3. Safety of Dalbavancin

Data on adverse drug reactions were recorded in the database, together with other relevant variables, including resistance development during treatment.

### 4.4. Statistical Analysis

Data were recorded in an anonymous electronic database. Qualitative variables were described using absolute and relative frequencies, and quantitative variables as mean ± standard deviation (SD) for normally distributed data or median with interquartile range (IQR) otherwise. The Chi-square test was used for analysis of categorical variables, while Student’s *t*-test and ANOVA were applied to compare categorical with quantitative variables, depending on the number of categories. Factors associated with treatment failure were identified through statistical analysis. Analyses were performed using Stata^®^ V.16.0.

All study data were collected anonymously, and the study was approved by the hospital’s Research Ethics Committee (code number 208/22). Due to the retrospective design of the study, informed consent was waived.

## 5. Conclusions

Dalbavancin appears to be a safe and convenient alternative for suppressive therapy in orthopaedic implant-related infections. Although resistance development was infrequent, it remains a concern and warrants close monitoring.

## Figures and Tables

**Figure 1 antibiotics-14-01171-f001:**
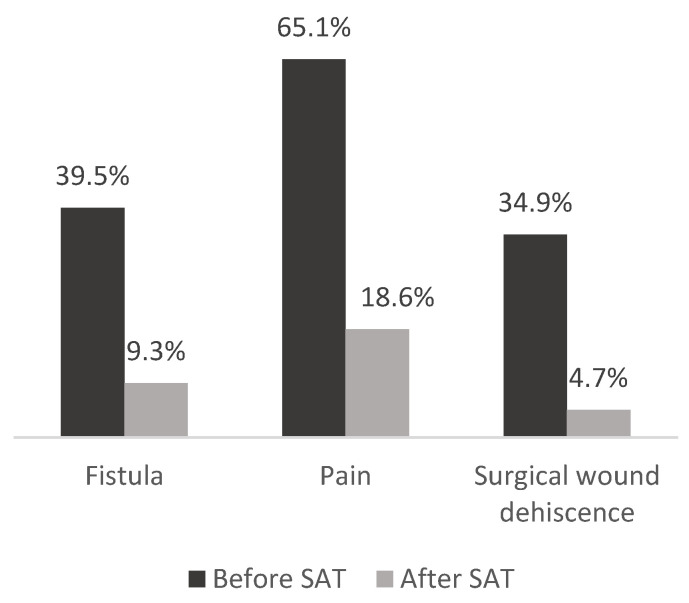
Symptoms of implant-related infection before and after SAT.

**Table 1 antibiotics-14-01171-t001:** Characteristics of implant-associated osteoarticular infections.

Characteristics	n (%)
Prosthetic joint infection (PJI)	38 (88.4)
Location: Knee	25 (65.8)
Hip	12 (31.6)
Shoulder	1 (2.6)
Revision prosthesis	21 (56.8)
Timing: Acute	12 (31.5)
Chronic	21 (55.2)
Hematogenous	2 (5.2)
PIOC	3 (7.8)
Osteosynthesis implant infection (OII)	3 (7.0)
Location: tibia	2 (66.7)
Femur	1 (33.3)
Timing: late (more than 10 weeks)	3 (100)
Vertebral instrumentation infection	2 (4.7)
Microorganisms isolated	
- *S. epidermidis*	19 (44.2)
- *S. aureus*	8 (18.6)
- *E. faecalis*	4 (9.3)
- *E. faecium*	3 (7.0)
- *C. acnes*	3 (7.0)
- *S. lugdunensis*	1 (2.3)
- Coagulase-negative *Staphylococcus*	1 (2.3)
- Negative cultures	3 (7.0)
Surgical management:	
- DAIR ^1^	21 (48.8)
- One stage-revision arthroplasty	3 (7.0)
- Implant removal ^1^	1 (2.3)
- No surgery	18 (41.9)

^1^ SAT was initiated due to early failure.

**Table 2 antibiotics-14-01171-t002:** Variables related to dalbavancin administration.

Characteristics of Dalbavancin Administration	n (%)
Treatment type: Targeted Empirical	40 (93.0) 3 (7.0)
Setting of first dose administration: Inpatient Outpatient clinic	25 (58.1) 18 (41.9)
Time from surgery to dalbavancin initiation (days)	118 (67–468)
Time from OAI diagnosis to dalbavancin initiation (days)	210 (54–1535)
Antibiotics used before dalbavancin:	41 (95.6)
Beta-lactams	12 (27.9)
Clindamycin	2 (2.3)
Quinolones	10 (23.3)
Vancomycin	11 (25.6)
Daptomycin	13 (30.2)
Cotrimoxazole	10 (23.3)
Rifampin	10 (23.3)
Linezolid	16 (37.2)
Tedizolid	6 (14.0)
Doxycycline	5 (11.6)
Reasons for choosing dalbavancin	
- To improve adherence	21 (48.8)
- Toxicity of other antibiotics	18 (41.9)
- Resistance to other antibiotics	12 (27.9)
- To shorten hospitalisation	9 (20.9)
- Potential drug–drug interactions	2 (4.7)
- Allergy to other antibiotics	1 (2.3)
First dalbavancin dose: 500 mg	2 (5.0)
1000 mg	16 (38.1)
1500 mg	24 (57.1)
Subsequent regimens:	
500 mg weekly	6 (14.0)
500 mg every 2 weeks	1 (2.3)
1000 mg weekly	3 (7.0)
1000 mg every 2 weeks	8 (18.6)
1000 mg every 3 weeks	1 (2.3)
1000 mg monthly	2 (4.7)
1500 mg every 2 weeks	11 (25.6)
1500 mg every 3 weeks	1 (2.3)
1500 mg monthly	10 (23.3)
Number of doses	25 (11–48)
Duration of SAT with dalbavancin (days)	333 (105–957)
Cumulative dalbavancin dose (mg)	30,000 (13,000–64,000)
Combination with other antibiotics	5 (11.6)

Quantitative variables are presented as median and interquartile range (IQR).

**Table 3 antibiotics-14-01171-t003:** Description of patients who developed resistance to dalbavancin.

Case	Characteristics	Duration of Dalbavancin (Days)	Cumulative Dose (mg)	Regimen (Inicial/Maintenance)	Time to Failure (Days)	Failure Management	MICs (Initial Strain) (mg/L)
2	Female, 37 y; chronic knee PJI; *S. haemolyticus*; no surgery	34	3000	1500 mg/ 1500 mg/every 3 wks	27	Switch of antibiotic, no surgery. Failure confirmed by new arthrocentesis due to persistent symptoms (MIC increased from 0.19 to 1)	V: 2 T: 4 Dap: ≤0.5 D: 0.19
5	Male, 78 y; chronic knee PJI; *S. epidermidis*; no surgery	506	34,000	1500 mg/ 1000 mg/every 2 wks	276	Loosening and pain, required 2-stage revision. Cultures showed two *S. epidermidis* strains (one susceptible, MIC 0.06; one resistant, MIC 1.5)	V: 2 T: 8 Dap:0.5 D: 0.06
9	Male, 69 y; acute knee PJI; *S. epidermidis*; DAIR	1053	85,500	1500 mg/ 1500 mg/every 2 wks	1053	Dosing interval extended to every 3 weeks; patient developed local signs and increased CRP. One-stage revision performed (explanted MIC = 2)	V:1 T:1 Dap: 0.25 D:0.012

**Table 4 antibiotics-14-01171-t004:** Events observed during SAT with dalbavancin.

Events	n (%)
Failure	11 (25.6%)
Development of resistance	3 (7.0%)
Adverse events	1 (2.3)
Mortality	5 (11.6)

## Data Availability

The original contributions presented in this study are included in the article/Appendix A. Further inquiries can be directed to the corresponding author.

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
