# Peer review of "Dalbavancin as Suppressive Therapy for Implant-Associated Osteoarticular Infections"

_antibiotics, 2025, doi:10.3390/antibiotics14111171_

Round 1
Reviewer 1 Report
Comments and Suggestions for Authors
Thank you for conducting this research. I have few points to be addressed to improve the manuscript:
1- What was the period of time covered by the study, and when was the retrospective work initiated. This should be clarified in the methods and if possible in the abstract.
2- The 2nd paragraph in the introduction needs to cite references.
3- The reasons for choosing dalbavancin needs to be clarified in the methods regarding the sources of the data that indicate or mention such reasons.
4- The methods should state the general sources of the data.
5- In table 1, it is not clear whether number of doses, duration of SAT, cumulative dose, and combined with other antibiotics were presented as (mean) or something else.
Author Response
1. What was the period of time covered by the study, and when was the retrospective work initiated? This should be clarified in the methods and, if possible, in the abstract.
Thank you for pointing this out. The information has been added to both the Methods section and the Abstract.
2. The 2nd paragraph in the introduction needs to cite references.
A reference to the Summary of Product Characteristics for dalbavancin has been added. The numbering of the remaining references has been updated accordingly.
3. The reasons for choosing dalbavancin need to be clarified in the methods regarding the sources of the data that indicate or mention such reasons.
As stated in the text, “The decision to use dalbavancin as suppressive therapy was made by the treating physician prior to study initiation” (line 263). Please let us know if you think further clarification would be helpful.
4. The methods should state the general sources of the data.
Thank you for this comment. This information has now been clarified in the Methods section.
5. In Table 1, it is not clear whether number of doses, duration of SAT, cumulative dose, and combined with other antibiotics were presented as (mean) or something else.
I believe this comment refers to Table 2. As stated in the table footnote: “Quantitative variables are presented as median and interquartile range (IQR).”
Reviewer 2 Report
Comments and Suggestions for Authors
Abstract:
There are discordant statements in the abstract section, analysis of variables associated with SAT failure gives more benefit to the reader. A long-acting antibiotic, dalbavancin, is prescribed whenever treatment failure with previous treatment (Ruiz-Sancho, 2023); unfavorable outcome or treatment failure of dalbavancin is associated with MRSA and the number of doses (Lovatti, 2023).
Page 1 lines 37-39: The aim of this study is to describe the characteristics and clinical outcomes of patients diagnosed with implant-associated osteoarticular infections receiving dalbavancin as a SAT.
Page 1 lines 43-44: Patient cohort characteristics were described, and an analysis was performed of variables associated with SAT failure under dalbavancin.
Results: A total of 43 patients
Reference:
Lovatti S, Tiecco G, Mulé A, Rossi L, Sforza A, Salvi M, Signorini L, Castelli F, Quiros-Roldan E. Dalbavancin in Bone and Joint Infections: A Systematic Review. Pharmaceuticals (Basel). 2023 Jul 15;16(7):1005. doi: 10.3390/ph16071005. PMID: 37513919; PMCID: PMC10385685.
Ruiz-Sancho A, Núñez-Núñez M, Castelo-Corral L, Martínez-Marcos FJ, Lois-Martínez N, Abdul-Aziz MH, Vinuesa-García D. Dalbavancin as suppressive antibiotic therapy in patients with prosthetic infections: efficacy and safety. Front Pharmacol. 2023 Jun 28;14:1185602. doi: 10.3389/fphar.2023.1185602. PMID: 37448966; PMCID: PMC10337584.
Introduction
The author can add mechanism of dalbavancin treatment failure.
Methods and Results
Could the author perform factor analysis for variables: Antibiotics used before dalbavancin, Reasons for choosing dalbavancin, and dalbavancin dose (in Table 1)? What is the physician's consideration before the Dalbavancin regimen (dosage and duration)?
Comments on the Quality of English Languagemoderate
Author Response
Thank you for taking the time to review our manuscript.
1) There are discordant statements in the abstract section, analysis of variables associated with SAT failure gives more benefit to the reader. A long-acting antibiotic, dalbavancin, is prescribed whenever treatment failure with previous treatment (Ruiz-Sancho, 2023); unfavorable outcome or treatment failure of dalbavancin is associated with MRSA and the number of doses (Lovatti, 2023).
Page 1 lines 37-39: The aim of this study is to describe the characteristics and clinical outcomes of patients diagnosed with implant-associated osteoarticular infections receiving dalbavancin as a SAT.
Page 1 lines 43-44: Patient cohort characteristics were described, and an analysis was performed of variables associated with SAT failure under dalbavancin.
Results: A total of 43 patients
Reference:
Lovatti S, Tiecco G, Mulé A, Rossi L, Sforza A, Salvi M, Signorini L, Castelli F, Quiros-Roldan E. Dalbavancin in Bone and Joint Infections: A Systematic Review. Pharmaceuticals (Basel). 2023 Jul 15;16(7):1005. doi: 10.3390/ph16071005. PMID: 37513919; PMCID: PMC10385685.
Ruiz-Sancho A, Núñez-Núñez M, Castelo-Corral L, Martínez-Marcos FJ, Lois-Martínez N, Abdul-Aziz MH, Vinuesa-García D. Dalbavancin as suppressive antibiotic therapy in patients with prosthetic infections: efficacy and safety. Front Pharmacol. 2023 Jun 28;14:1185602. doi: 10.3389/fphar.2023.1185602. PMID: 37448966; PMCID: PMC10337584.
--> Thank you for suggesting these references. I have included the study by Lovatti. The Ruiz-Sancho reference was already included, so I have kept it. I have also made some changes in the abstract to avoid inconsistencies and to include the analysis of SAT failure as a secondary objective.
2) Introduction
The author can add mechanism of dalbavancin treatment failure.
--> Thank you for pointing this out. A sentence with the appropriate references has been added.
3) Methods and Results
Could the author perform factor analysis for variables: Antibiotics used before dalbavancin, Reasons for choosing dalbavancin, and dalbavancin dose (in Table 1)? What is the physician's consideration before the Dalbavancin regimen (dosage and duration)?
--> As this was a retrospective study, the decision to use dalbavancin as suppressive antibiotic therapy was made by the treating physician.
The antibiotics used before dalbavancin, the reasons for its selection, and the dosing regimens are shown in Table 2.
Could you please clarify whether you are suggesting that both tables should be combined?
Reviewer 3 Report
Comments and Suggestions for Authors
Thank you for your effort on the paper.
My comments:
The introduction section is ok. The authors clearly presented the aim of the study.
In the results section. There is an inconsistency in the total number of acute and chronic cases. The authors said there were 29 chronic and 11 acute cases, yet they said they had 43 patients in total. Please correct the inconsistency.
Was there any relation with the dosage and treatment failure or complications?
The discussion section is well-written.
In the materials and methods section, it could be better to describe that the dose selection of the physicians was based on what?
The references are ok.
The tables and the figures are ok.
Author Response
Thank you for your revision. Following I attached the answer to your comments.
In the results section. There is an inconsistency in the total number of acute and chronic cases. The authors said there were 29 chronic and 11 acute cases, yet they said they had 43 patients in total. Please correct the inconsistency.
->Thank you for pointing out the mistake. I have corrected it accordingly.
Was there any relation with the dosage and treatment failure or complications?
-> No, there was not. Supplementary table 2 describes the variables associated with failure and includes the univariable analysis.
In the materials and methods section, it could be better to describe that the dose selection of the physicians was based on what?
-> I have clarified in the text that the choice of dalbavancin and the dosage were determined by the treating physician prior to the start of the study.
Reviewer 4 Report
Comments and Suggestions for Authors
General Comments on the Manuscript:
In this multicentre, observational study, the authors described the characteristics and clinical outcomes of patients diagnosed with implant-associated osteoarticular infections receiving dalbavancin as a Suppressive antibiotic therapy. Below are my specific comments and suggestions for improvement:
- Introduction: (Line 73)
The sentence: “Dalbavancin is generally used off-label, beyond its approved”, please add before it, what is the on-label indication of dalbavancin. - Methods – Inclusion Criteria and Protocol:
Please clearly specify the inclusion and exclusion criteria, end points as well as experimental protocol. For clarity and transparency, consider presenting the inclusion and exclusion criteria in a separate, clearly labeled subsection. - Methods: adverse drug reactions:
This section is missing please add the method used to evaluate the development of the ADRs
- Statistical Analysis:
It is missing
- Results – Clinical Characteristics:
Please include more detailed clinical characteristics of the study population. Specifically, indicate whether participants had conditions such as obesity, diabetes, autoimmune disorders, or other relevant comorbidities. Moreover, indicate age, sex and gender characteristics. - Results – Table 1 and 2:
Please separate in column number from percentage and describe the results respect to age and sex. - Results – Figure 1:
Please revise it. It is not easy to understand and separate the data respect to sex and gender. Moreover separate pain respect to the type and intensity. - Results – Drug Use / Polypharmacy:
Include information on medication use or polypharmacy, and explain if this could significantly influence the effect of dalbavancin. - Results – Table 3:
Please explain why the patients performed a continuative treatment with dalbavancin for 506 (patient 5) and 1053 days (patient 9), is it an on-label time? - Results:
Please describe the same characteristics reported in table 3 also for the other patients, in order to evaluate if the type of microorganism, infection, time of treatment, dosage, have or not influenced the effect of the treatment.
- Results:
Please describe the type of ADRs observed and how you have treated these.
- Discussion:
Please revise the Discussion to better integrate and reflect the results presented. Ensure that each major finding is discussed in terms of its significance, implications, and potential mechanisms, and compare your findings with those of previous studies.
Author Response
- Introduction: (Line 73)
The sentence: “Dalbavancin is generally used off-label, beyond its approved”, please add before it, what is the on-label indication of dalbavancin.
->The text said: "beyond its approved indication for skin and soft tissue infections". A general problem in the treatment of osteoarticular infections for several decades has been that no new antibiotics have been registered for the treatment of osteomyelitis or implant-associated infections. This tends to limit the rate at which they are incorporated into medical practice in certain situations where they may be most useful.
- Methods – Inclusion Criteria and Protocol:
Please clearly specify the inclusion and exclusion criteria, end points as well as experimental protocol. For clarity and transparency, consider presenting the inclusion and exclusion criteria in a separate, clearly labeled subsection.
--> Thank you for your comment. We have reorganised the methods section for greater clarity.
- Methods: adverse drug reactions: This section is missing please add the method used to evaluate the development of the ADRs
-->Thank you for your comment. We have added how adverse events were collected.
- Statistical Analysis:
It is missing --> The manuscript includes the statistical methods used. In the new version, they are specified in a sub-section.
- Results – Clinical Characteristics:
Please include more detailed clinical characteristics of the study population. Specifically, indicate whether participants had conditions such as obesity, diabetes, autoimmune disorders, or other relevant comorbidities. Moreover, indicate age, sex and gender characteristics
--> As this was a retrospective study, not all comorbidities were captured in the electronic database. However, age, sex, and clinically relevant comorbidities related to the study were recorded as can be seen at the beginning of the results section
- Results – Table 1 and 2:
Please separate in column number from percentage and describe the results respect to age and sex
-->I have no objection to making this modification if necessary. However, I have noticed that in other articles published in the journal, the data in the tables are presented in the same column format, Adding more columns with age and gender information to these tables would complicate their visualisation. If the editor deems it necessary, we can modify them
Results – Figure 1:
Please revise it. It is not easy to understand and separate the data respect to sex and gender. Moreover separate pain respect to the type and intensity.
--> The aim of the graph is to show how many patients presented these signs (presence of fistula, pain, or wound dehiscence) before starting suppressive therapy with dalbavancin and how they evolved by the end of follow-up. We believe that age and sex do not have an impact on the purpose of this figure.
The point about pain intensity is interesting. Unfortunately, this variable was not collected in a quantitative manner.
- Results – Drug Use / Polypharmacy:
Include information on medication use or polypharmacy, and explain if this could significantly influence the effect of dalbavancin.
--> This information was not recorded in our database.
- Results – Table 3:
Please explain why the patients performed a continuative treatment with dalbavancin for 506 (patient 5) and 1053 days (patient 9), is it an on-label time?
--> this cohort describes the use of dalbavancin for an off-label indication, specifically the treatment of osteoarticular infections. The purpose of this therapy is symptomatic control; therefore, its duration is indefinite.
- Results:
Please describe the same characteristics reported in table 3 also for the other patients, in order to evaluate if the type of microorganism, infection, time of treatment, dosage, have or not influenced the effect of the treatment.
-->The inclusion of the rest of the patients in this table would make it unmanageable. We thought it would be interesting to show the characteristics of the patients who developed resistance. As can be seen, two of them had elevated MICs for teicoplanin, and the doses and dosing intervals for dalbavancin do not suggest underdosing.
- Results:
Please describe the type of ADRs observed and how you have treated these.
-->This information is described in lines 132–135.
- Discussion:
Please revise the Discussion to better integrate and reflect the results presented. Ensure that each major finding is discussed in terms of its significance, implications, and potential mechanisms, and compare your findings with those of previous studies
--> We have revised and reorder the discussion section. Thank you for your comments.
Round 2
Reviewer 4 Report
Comments and Suggestions for Authors
Dear Authors,
I have read the manuscript that has ben partially improved. Please add the limitations